# Bio-Inspired 4D Printing of Dynamic Spider Silks

**DOI:** 10.3390/polym14102069

**Published:** 2022-05-19

**Authors:** Guiwei Li, Qi Tian, Wenzheng Wu, Shida Yang, Qian Wu, Yihang Zhao, Jiaqing Wang, Xueli Zhou, Kunyang Wang, Luquan Ren, Ji Zhao, Qingping Liu

**Affiliations:** 1Advanced Materials Additive Manufacturing ((AM)2) Lab, School of Mechanical and Aerospace Engineering, Jilin University, Changchun 130025, China; ligw@jlu.edu.cn (G.L.); tianqi1819@mails.jlu.edu.cn (Q.T.); wzwu@jlu.edu (W.W.); yangsd9919@mails.jlu.edu.cn (S.Y.); zhaoyh9919@mails.jlu.edu.cn (Y.Z.); wjq9919@mails.jlu.edu.cn (J.W.); lqren@jlu.edu.cn (L.R.); jzhao@jlu.edu.cn (J.Z.); 2Key Laboratory of Bionic Engineering (Ministry of Education), Jilin University, Changchun 130022, China; wuqian21@mails.jlu.edu.cn (Q.W.); xlzhou@jlu.edu.cn (X.Z.); kywang@jlu.edu.cn (K.W.); 3School of Mechanical Engineering and Automation, Northeastern University, Shenyang 110004, China

**Keywords:** 4D printing, bio-inspired spider silks, adjustable mechanical properties, shape morphing, stimulus response

## Abstract

Spider silks exhibit excellent mechanical properties and have promising application prospects in engineering fields. Because natural spider silk fibers cannot be manufactured on a large scale, researchers have attempted to fabricate bio-inspired spider silks. However, the fabrication of bio-inspired spider silks with dynamically tunable mechanical properties and stimulation–response characteristics remains a challenge. Herein, the 4D printing of shape memory polyurethane is employed to produce dynamic bio-inspired spider silks. The bio-inspired spider silks have two types of energy-absorbing units that can be adjusted, one by means of 4D printing with predefined nodes, and the other through different stimulation methods to make the bio-inspired spider silks contract and undergo spiral deformation. The shape morphing behaviors of bio-inspired spider silks are programmed via pre-stress assemblies enabled by 4D printing. The energy-absorbing units of bio-inspired spider silks can be dynamically adjusted owing to stress release generated with the stimuli of temperature or humidity. Therefore, the mechanical properties of bio-inspired spider silks can be controlled to change dynamically. This can further help in developing applications of bio-inspired spider silks in engineering fields with dynamic changes of environment.

## 1. Introduction

The spider web has excellent mechanical properties [1,2,3], combines high tensile strength with high ductility, and simultaneously exhibits excellent impact resistance [4,5]. The tensile strength of silk fibers is comparable to that of steel, and the elasticity of natural spider silk is almost the same as that of rubber [6]. Spider silk has good application prospects in various fields, such as textiles [7,8], biomedicine [9,10], military affairs [11], and aerospace [12]. However, spiders are carnivores—they do not like to live in groups and even kill each other—rendering it difficult for spiders to be bred on a large scale [13,14]. Therefore, producing natural spider fibers on an industrial scale is challenging. Thus, the manufacturing and molding of bio-inspired spider silk has become a popular research topic.

Numerous studies have focused on the manufacture of bio-inspired spider webs. Vendrely et al., through genetic engineering methods, employed two biotechnological production strategies to successfully apply spider silk protein for the production of spider silk [15]. Venkatesan et al. fabricated engineered major ampullate spidroin 2 (eMaSp2) fiber by introducing foreign N- and C-terminal domains, and demonstrated that the fiber has good shape memory effects triggered by humidity factors and the ability to restore stress [16]. Inspired by the microstructure of spider silk, Chan et al. effectively utilized the localized β-sheet domains in the amorphous network considering the shortcomings of existing processing technology, and synthesized a super composite material with a spider silk-like “amorphous/β-folding” microstructure [17]. Dou et al. self-assembled homemade hydrogel fibers to create artificial spider silk. This faux spider silk fiber consists of a plastic sheath placed around an elastic core, which results in spider silk-like strength and stretchability [18]. When used in impact reduction applications, the bio-inspired spider silk fiber has a negligible rebound, allowing it to be applied to kinetic energy cushioning and shock reduction. Zou et al. successfully designed a transparent composite material with super-impact resistance using the SBHL strategy. Zou et al.’s biologically inspired spider webs are able to dissipate a significant amount of energy, as they absorb impact and seize the projectile like a natural spider web [19]. Qin et al. created spider-web mimics composed of elastic thin filaments and investigated the mechanical response of the elastic web under a variety of loading conditions [20].

The studies discussed above focused on static bio-inspired spider silk to create artificial bio-inspired fibers with mechanical properties similar to those of natural spider silk. However, natural spider silk has a hierarchical structure and multiple properties, such as humidity/water responses, light transmittance, and shape memory [21]. The actual application conditions for bio-inspired spider silk often change dynamically. For the impact resistance of spider silk it is often necessary to adapt to dynamic adjustments. We used 4D printing to create a spider silk structure. Based on 3D printing, 4D printing technology has an additional fourth dimension: the time dimension [22,23]. Thus, 4D-printed objects can change their shape/structure and mechanical properties over time after being stimulated [24,25,26]. Therefore, bio-inspired spider silk obtained by 4D printing can be stimulated by different conditions, such as temperature and humidity, to achieve dynamic changes. By controlling the difference between the stimulus and preset printing conditions, the mechanical properties of the bio-inspired spider silk can be controlled to change dynamically, which can widen the application prospects of bio-inspired spider silk in the engineering field.

## 2. Materials and Methods

### 2.1. Mechanical Simulation Analysis

We chose CATIA V5R21 modeling software to establish the spider silk model and used Workbench 2021 R1 software for the finite element simulation analysis of the model. In the finite element analysis, one end of the model was set as a fixed support and the other end was loaded with a tension of 1 N. For the material setting of the model, the density was set to 1250 kg/m^3^, the Young‘s modulus was 1.4 MPa, and the ultimate tensile strength was 10 MPa. This solved the equivalent effect diagram of the model.

### 2.2. Printing the Bio-Inspired Spider Silk Samples

The raw material selected for printing was polyurethane (SK96B09, Kyoraku Co., Ltd., Tokyo, Japan). For the 4D printing of bio-inspired spider silk, the following process parameters were set: the bottom layer printing speed was 300 mm/min, the top layer printing speed was 7000 mm/min, the layer height was 0.2 mm, the nozzle diameter was 0.4 mm, the extrusion magnification was 1.2, the extrusion line width was 0.4 mm, the print nozzle temperature was 180 °C, and the substrate temperature was 20 °C. This printed the required bio-inspired spider silk with different nodes. At the same time, more than 10%, 30%, 50%, and 70% filling densities were employed.

### 2.3. Mechanical Property Characterization Experiment

For the mechanical property characterization experiments, the electronic universal testing machine (UTM6104, Shenzhen Sansi Longitudinal and Hori-zontal Technology Co., Ltd., Shenzhen, China) was used. We mainly conducted tensile property characterization experiments on bio-inspired spider silk to investigate the effects of tensile speed, stimulus time, and spider silk humidity on the tensile properties of bio-inspired spider silk. First, we investigated the tensile properties of the unstimulated sample and the samples stimulated for 1 min at 60 °C with different densities of nodes at a tensile speed of 30 mm/min. Second, we investigated the tensile properties of the unstimulated sample and the samples stimulated for 2 min at 60 °C with the same density of nodes at different tensile speeds. Finally, the tensile properties of bio-inspired spider silks with different stimulation times were investigated at a tensile speed of 30 mm/min. A universal mechanical testing machine was used to test the tensile properties of bio-inspired spider silk at a constant tensile speed until the specimens were pulled off. The stretching speed was chosen to be 30 mm/min and 30, 50, 70, 90, 150, and 210 mm/min when the effects of different stretching speeds were explored. For each set of tensile property characterization experiments, at least three mechanical samples were tested to determine the yield point and the corresponding elastic limit force, as well as the fracture point and the corresponding maximum tensile force. The elongation at the break of the bio-inspired spider was obtained by dividing the value of the deformation displacement at the break by the original length of the sample. The average value was recorded, and the standard deviation was marked on the histogram. Then, the sample test result closest to the average value was selected, according to which the force–tension displacement curve of the bio-inspired spider was drawn in the same way.

### 2.4. The DSC and TGA Experiments

The DSC (DSC25, TA Instruments, New Castle, DE, USA) experiments were conducted to test the difference in the degree of glass transition of bio-inspired spider samples with different moisture absorption, and the temperature rise rate was set to 5 °C/min from 0 to 200 °C. The TGA (TGA55, TA Instruments, New Castle, DE, USA) experiments were conducted to verify the moisture content absorbed by the bio-inspired spider silk sample after immersion for different times. The immersion times were chosen to be 4, 8, and 16 min at 60 °C and 30 and 60 min at 40 °C. The heating rate of the TGA experiment was 5 °C/min in the range of 30 °C to 280 °C.

### 2.5. Bio-Inspired Spider Web Stimulation Experiments

The webs were stimulated with 150 W infrared light from a distance of 35 mm to observe the shrinkage and deformation of both the unstimulated webs and high-temperature-stimulated webs. The infrared light (MYK020, Shanghai Yafu Lighting Electric Co., Ltd., Shanghai, China) was equipped with a Philips infrared bulb (PAR38E 150W 230V, Amsterdam, The Netherlands). Next, infrared light and humidity were used to stimulate the high-temperature-stimulated webs to compare the differences between the two stimulation methods. The high-temperature stimulation of the bio-inspired spider silk was performed using water at a constant temperature of 80 °C. Simulations were performed three times; each time, after 20 s of stimulation, the bio-inspired spider silk was quickly removed, gently straightened, and stimulated again three times. The humidity stimulation method involved soaking and humidifying the bio-inspired spider silk with constant-temperature water at 25 °C.

### 2.6. Impact Resistance Experiment and Invisible Interception Experiment

Application experiments were also conducted, including an impact resistance experiment. Bio-inspired spider webs 100 × 100 mm in size were spaced 12.5 mm apart, and the impact experiments were conducted using a glass ball and tennis ball. During the impact test, it was ensured that the test was conducted under windless conditions. The vertical distance between the center of the ball and the net surface was 30 mm for both balls. No additional force was attached to the ball during its fall, and the impact was taken in a free-fall manner. The diameter of the glass balls was 25 mm, and the weight of the tennis balls was 55.638 g. Photographs were taken using an industrial camera (OSG030-815UM, Shenzhen Yingshi Technology Co., Ltd., Shenzhen, China) equipped with a 2 million 1/2” target surface industrial lens (HF-Z0412, Shenzhen Yingshi Technology Co., Ltd., Shenzhen, China) to demonstrate the feasibility of the bio-inspired spider silk applied to stealth interception. The dynamic bio-inspired spider web was used to demonstrate the feasibility of intercepting impacts.

Next, bio-inspired spider web stealth interception experiments were performed, in which the same background was chosen and spider silk with different stimulation levels was used to weave the web; the difference in the stealthiness of the bio-inspired spider web was observed under different stimulation levels, and the feasibility of its application to stealth interception was analyzed. The bio-inspired spider silk was stimulated once, three times, and five times at a constant temperature of 80 °C. The samples were removed quickly after each stimulation for 20 s and gently straightened so that the corresponding number of repetitions could obtain three types of bio-inspired spider silk with different stimulation levels. Notably, bio-inspired spider silk was removed more slowly and not straightened during the fifth stimulation to maintain the spiral effect.

## 3. Results and Discussion

### 3.1. Bio-Inspired 4D Printing of Dynamic Spider Silks

A biological prototype of spider silk is shown in Figure 1a. It is both elastic and rigid, and can be considered to be composed of two parts: a semi-amorphous domain and a nanocrystal [19]. When bio-inspired spider silk is stretched by force, the semi-amorphous domain part inside the bio-inspired spider silk becomes elastic, and the spiral part inside it is stretched by force. It can absorb energy when the bio-inspired spider silk is stressed, which provides the bio-inspired spider silk with good impact resistance. Based on this bio-inspired principle, as shown in Figure 1b, we programmed the process parameters. It is possible to print two layers at different speeds and then bury the preset nodes. Under stimulus, the linear part in the middle is spirally deformed owing to the difference in the shrinkage of the double-layer material, while the preset nodes retain their bending at the nodes so that a double preset can be achieved. The design enables the bio-inspired spider silk to achieve spiral deformation, as shown on the right side of Figure 1b, under certain stimulation conditions.

Stimulation experiments were performed on the printed spider silk with different nodes. The stimulation experiments were performed under the specific conditions mentioned in Figure 1c, where bio-inspired spider silk with different nodes was placed in constant-temperature water for the same amount of time. The time selected was 1 min, the temperature gradient was 10 °C, and the temperature range was 40 to 80 °C. The density of nodes of the bio-inspired spider silk differed; 100 × 2 indicates that the bio-inspired spider silk had a total of two segments, each with a length of 100 mm. The preset node 50 × 4 means that the bio-inspired spider silk consisted of four segments, each with a length of 50 mm and three preset nodes. Thus, under the premise that the density of the nodes of the bio-inspired spider silk differs, the total length of the selected segments is the same (200 mm), and the influence of different node densities and different temperatures on the stimulation of the bio-inspired spider silk can be observed. Next, mechanical property characterization experiments were performed. The force simulation of the bio-inspired spider silk is shown in Figure 1d; the figure shows that when the printed spider silk with nodes is stretched by force, because the equivalent force at the node is significantly greater than that at a smooth straight line, the node will lead the force expansion to offset part of the pull impact, as shown in Figure 1(d_1_). Similarly, the mechanical simulation of the stimulated spider silk is shown in Figure 1(d_2_); at this time, because the linear part in the middle of the original spider silk is bent spirally, the maximum equivalent force appears at the relatively uniform spiral. When the bio-inspired spider silk is stimulated and stretched by the force, the spiral part first unfolds due to the stretching force to offset part of the impact, and then the nodes are expanded by the force.

We prepared the bio-inspired spider silk with different nodes by pre-programming the process parameters. As shown in Figure 1c, the higher the temperature during stimulation, the more pronounced the spiral bending phenomenon of the bio-inspired spider silk. On the basis of the original bending of the nodes being retained, the linear part of the printed bio-inspired spider silk also undergoes spiral deformation. Additionally, when the bio-inspired spider silk is subjected to tensile deformation, the spiral deformation region generated by the linear part will be the first to be subjected to a larger value of equivalent force and unfold to resist the tensile impact. In this way, together with the bending at the original nodes, a double pre-setting is achieved to enhance the tensile impact resistance of the bio-inspired spider silk. This method uses a double-preset approach to provide a better energy-absorption effect.

### 3.2. Effect of Energy-Absorbing Units on the Tensile Properties of Bio-Inspired Spider Silks

The size and density of energy-absorbing units affect the stress distribution of the bio-inspired spider silk sample during stretching, which affects the dynamics of the mechanical properties of the sample under different conditions. Two types of energy absorption units exist in the samples. The density and size of the nodes are preset when printing the bio-inspired spider silk sample as the preset energy-absorbing unit. The spiral deformation of the sample after stimulation can be regarded as a large node as the additional energy absorption unit. The energy-absorbing unit is mainly controlled by temperature during stimulation, which can control the degree of spiral deformation. Thereby, the influences of the density, the size of nodes, and the stimulation temperature on the tensile properties of the bio-inspired spider silk samples were investigated.

#### 3.2.1. Effect of Node Density

The density of the nodes significantly affected the tensile properties of the bio-inspired spider silk. Four types of spider silk with different densities of nodes were used to conduct tensile experiments under both unstimulated and stimulated conditions to investigate specific difference phenomena. Figure 2a–d show the stretching of the four unstimulated raw silk materials. As shown in Figure 1(d_1_), the mechanical simulation results of the bio-inspired spider silk sample showed that when the raw bio-inspired spider silk sample was stretched by force, the equivalent force value at the node was large, and the first to be damaged. When the density of nodes was small, for example, in the raw bio-inspired spider silk sample of 100 × 2 with only one preset node, the stress was concentrated which first damaged the node, and the elastic limit force of the bio-inspired spider silk was small. When the density of the nodes was large, for example, the raw bio-inspired spider silk of 50 × 4 and 25 × 8, which had three and seven nodes to share the corresponding tensile stress, respectively, the stress first damaged these nodes. However, when the density of the nodes was large for the 12.5 × 16 spider silk sample with 15 nodes, the stress concentration area was too dense, affecting the straight section; thus, the elastic tensile limit and maximum tensile force of the unstimulated sample were reduced.

The unstimulated biomimetic silk had only the first type of energy absorbing units with the preset nodes. When the sample was subjected to tension, the nodes were subjected to a large value of equivalent force. However, the stress concentration at the nodes also affected the linear part. When the linear section was short (e.g., 12.5 mm), the elastic tensile limit and the maximum tensile force of the unstimulated sample was reduced. Therefore, adopting appropriate nodal densities (e.g., 50 × 4 and 25 × 8) can improve the tensile mechanical properties of bio-inspired spider silk.

We further investigated the differences in the tensile properties of the stimulated bio-inspired spider silk, as shown in Figure 2e–h. The elongation at the break of the bio-inspired spiders with different densities of nodes after stimulation remained the same, and there was no significant difference in the maximum tensile force value; only the elastic limit force value increased with a reduction in the density of nodes of the raw bio-inspired spider material. This is because a smaller density of nodes results in a longer linear segment, as shown in Figure 1(d_2_), and the equivalent maximum stress in the bio-inspired spider silk after stimulation when it is stretched by force occurs at the bend formed by the contraction of the linear segment after stimulation. In contrast, the 100 × 2 bio-inspired spider silk had the longest original straight segment and the most bending spirals formed at the same conditioned stimulation; thus, the elastic limit force of the bio-inspired spider silk with the lowest density of these node became the largest. This shows that the difference in the density of nodes causes a significant difference in the tensile properties of bio-inspired spider silks.

Two types of energy absorbing units are present in the stimulated bio-inspired spider silk sample. The spiral curved area formed by the stimulated bio-inspired spider silk sample acted as the second type of energy absorbing unit and was the first to be stretched by the force. Bio-inspired spider silks with different node densities have different numbers of the first type of energy absorption units—preset nodes—and they have different lengths of linear regions, which form the second type of energy-absorption units. The combined effect of the two types of energy-absorbing units leads to significant differences in the tensile properties of the stimulated bio-inspired spider silk samples.

#### 3.2.2. Effect of Node Size

In the previous section, we showed that the densities of nodes significantly affected the tensile properties of bio-inspired spider silk, and the size of the nodes should also affect the tensile properties of bio-inspired spider silk. The filling density parameter during printing was set to 10%, 30%, 50%, and 70% to obtain four types of bio-inspired spider silk structures with different node sizes. Among them, the bio-inspired spider silk node was the largest at the 10% filling density and the smallest at the 70% filling density. As shown in Figure 3a–d, tensile experiments were performed on unstimulated raw bio-inspired spider silk materials with different filling densities. The nodes of the samples printed at 70% filling density were the smallest, and extremely small nodes lead to more obvious stress concentration, which causes the bio-inspired spider silk to fracture in the middle of stretching; the elongation at the breakage of this bio-inspired silk sample was the smallest. The maximum tensile force of the unstimulated biomimetic 4D-printed bio-inspired spider silk sample at this point decreased with the size of the nodes, as shown in Figure 3c, which could also be because smaller nodes lead to a more pronounced stress concentration phenomenon which, in turn, reduces the maximum tensile force. Next, the raw bio-inspired spider silk material with different filling densities was stimulated and subjected to tensile experiments, as shown in Figure 3e–h. At this point, the bio-inspired spider silk had two types of energy-absorbing units after stimulation. The analysis performed under this condition is shown in Figure 3f. The elastic limit force of the bio-inspired spider silk sample with 70% filling density was slightly smaller, which may have been caused by the small nodes and obvious stress concentration. The fracture tensile rate of the bio-inspired spider silk sample under this condition decreased with a reduction in node size; the value of elongation at the break of the sample printed at 70% filling density was also the smallest.

The size of the nodes also significantly affects the tensile properties of the bio-inspired spider silk samples. When the nodes are small, the stress concentration is extremely pronounced and the degree of bending at these nodes is extremely dramatic. Thus, the smaller the node, the more likely it is to fracture due to excessive stress concentration. This results in relatively smaller values of maximum tensile force and fracture elongation for bio-inspired spider silk samples with smaller nodes.

#### 3.2.3. Effect of Stimulation Temperature

Finally, the effect of stimulation temperature on the tensile properties of the bio-inspired spider silk samples was investigated, as shown in Figure 4a–d. The elastic limit force, maximum tensile force, and elongation at the breaks of the samples were not significantly different when the stimulation temperature was relatively low, e.g., at 40, 50, or 60 °C for the constant-temperature water stimulation. When the temperature was again increased to 70 °C, the elastic limit force and elongation at the break of the stimulated bio-inspired spider samples significantly reduced, while their maximum tensile force magnitude remained essentially unchanged. When the temperature was raised to 80 °C again, the elastic limit force, maximum tensile force, and elongation at the break increased compared with those at 70 °C. The elastic limit force and the elongation at the break were lower than those of the bio-inspired spider samples stimulated by constant-temperature water at 40, 50, and 60 °C, but the maximum tensile force remained the same. The influence of stimulation temperature on the tensile properties of bio-inspired spider samples cannot be ignored.

We synthesized the effects of node density and size as well as stimulation temperature on the tensile properties of bio-inspired spider silk samples. Different stimulation temperatures influenced the differences in the large nodes formed with the bio-inspired spider silk samples after helical deformation. At the nodes, the abrupt change in shape produced the stress concentration phenomenon. Differences in the type, density and size of the nodes produced differences in the stress concentration, which further influenced the bio-inspired spider silk tensile properties.

### 3.3. Effect of Tensile Speed and Moisture Absorption Content on the Tensile Properties of the Bio-Inspired Spider Silks

#### 3.3.1. The DSC and TGA Results

The effects of water on the internal molecular characteristics of polyurethane were investigated via the DSC and TGA experiments. The glass transition temperature of polyurethane decreases with increases in water absorption [27]. The absorbed water can interact with the polymer, thus increasing the mobility of the polymer chains and leading to a decrease in the glass transition temperature [28].

Figure 5a–d show the DSC and TGA results. Figure 5a,c show the DSC results. Figure 5a illustrates that the glass transition temperatures of the bio-inspired spider samples soaked at 60 °C for 4, 8, and 16 min decreased with increasing soaking time. The glass transition temperature of the bio-inspired spider sample soaked for 16 min was the lowest, followed by that of the bio-inspired spider sample soaked for 8 min, and the glass transition temperature of the bio-inspired spider sample soaked for 4 min was the highest. Figure 5c illustrates the DSC experiments performed on the bio-inspired spider silk samples treated by immersion for 30 and 60 min at 40 °C. The glass transition temperature of the bio-inspired spider silk samples soaked for 30 min was higher than that of the bio-inspired spider silk samples soaked for 60 min. Figure 5b,d show the TGA results. Figure 5b shows that the moisture absorption of the bio-inspired spider silk samples soaked for 4, 8, and 16 min at 60 °C increased with increasing soaking time. The bio-inspired spider silk samples soaked for 16 min absorbed the most moisture, followed by the bio-inspired spider silk samples soaked for 8 min, and the bio-inspired spider silk samples soaked for 4 min absorbed the least moisture. Figure 5d illustrates the TGA experiments performed on the bio-inspired spider silk samples soaked at 40 °C for 30 and 60 min. The degree of moisture absorption was greater for the bio-inspired spider silk samples soaked for 60 min and lower for the bio-inspired spider silk samples soaked for 30 min. The increase in moisture absorption significantly reduced the glass transition temperature of the bio-inspired spider silk sample.

This is because the degree of moisture absorption of the bio-inspired spider silk samples can significantly affect their glass transition temperature. Therefore, we conducted experiments to investigate the effect of the degree of moisture absorption on the mechanical and tensile properties of spider silk. The magnitude of the impact of the bio-inspired spider silk samples varied by varying the stretching speed, representing the impact size of the bio-inspired spider silk and, by controlling the soaking stimulation time at 60 °C, the degree of moisture absorption of the samples could be varied.

#### 3.3.2. Effect of Different Stretching Speeds

Figure 6a–d show the unstimulated raw bio-inspired spider silk samples at different stretching speeds. At lower stretching speeds, that is, 30, 60, and 90 mm/min, the stretching profiles of this photo-bio-inspired spider silk material were similar, and the tensile properties of the bio-inspired spider silk were approximately the same. At higher stretching speeds, up to 110 mm/min, a significant reduction in the elongation at the break of the original spider silk material was observed. This phenomenon was more pronounced when the stretching speed was higher (150 mm/min). Thus, when the tensile properties of the 4D-printed samples were directly characterized without stimulation, the high tensile speed indicated that the bio-inspired spider silk samples were subjected to a high impact. This led to a fracture in the bio-inspired spider silk sample without reaching the desired tensile strength limit, resulting in a significant reduction in the elongation at the break. To avoid this phenomenon, we performed tensile property characterization experiments on bio-inspired spider silk samples after deformation by stimulation under certain conditions. As shown in Figure 6e–h, the stimulation conditions were chosen to allow the bio-inspired spider silk sample to absorb moisture by submerging it in water at a constant temperature of 60 °C for 2 min. The sample did not break in the middle of the experiment, even when the tensile speed was increased to 210 mm/min. The elastic limit force, maximum tensile force, and elongation at the break of the samples remained the same for different tensile speeds, and the tensile properties of the samples did not significantly vary from 30 to 210 mm/min.

Unstimulated bio-inspired spider silk samples were subjected to larger tensile speeds, which resulted in the sudden breakage of the bio-inspired spider silk samples during stretching, causing a significant decrease in the fracture elongation of the samples at the higher tensile speeds. In order to avoid mid-rupture, the sample can be soaked to increase the relative humidity of the sample so that the tensile speed is increased from the value that causes the unstimulated spider silk sample to break, and the stimulated sample still does not break mid-rupture. Thus, by soaking and humidifying the samples, breakage in the middle of the process at a high stretching speed can be prevented.

#### 3.3.3. Effect of the Degree of Moisture Absorption

We investigated the effect of the degree of moisture absorption on the tensile properties of the bio-inspired spider silk, as shown in Figure 7a–d, by subjecting the bio-inspired spider silk to immersion stimulation at 60 °C for 0, 1, 2, 4, 8, and 16 min, and then investigated the differences in the tensile properties of the samples under such conditions. The maximum tensile force of the samples with different degrees of moisture absorption remained the same, and there was no obvious difference; however, the curve of force versus tensile displacement of the sample with the greatest degree of moisture absorption, that is, the sample soaked for 16 min, significantly changed. The elastic limit force of the sample decreased significantly. In addition, this sample had the smallest elongation at its break compared to the previous samples that were soaked for 0, 1, 2, 4, and 8 min. The difference in soaking stimulation time significantly affected the elastic limit force and elongation at the break of the bio-inspired spider silk samples.

When the stimulated bio-inspired spider silk samples were immersed for longer periods of time, the elastic ultimate force and elongation at the break decreased as the degree of moisture absorption of the bio-inspired spider silk samples increased. The tensile strength of the polymer decreased with increasing water absorption. This is related to the increase in toughness of the bio-inspired spider samples after a large increase in the degree of moisture absorption of the bio-inspired spider silk.

### 3.4. Bio-Inspired Spider Web Stimulation Experiment

Different webs undergo relatively regular dynamic changes when stimulated in various ways. Figure 8 shows the deformation diagrams of the bio-inspired spider silk webs prepared with unstimulated and high-temperature-stimulated spider silk. Figure 8a shows the dynamic deformation process of a web woven with unstimulated spider silk. Figure 8b shows the deformation process of the web woven by the bio-inspired spider silk stimulated at high temperatures. Dynamic changes were observed in both webs under infrared light stimulation. The overall deformation is a contraction process. Thus, the response rate of the webs woven by samples stimulated at high temperatures is significantly better than that of the webs woven by unstimulated spider silk when subjected to the stimulation. This is related to the learning property of the stimulated response of the shape memory polymer [29].

Figure 8c,d show the results for the square-prepared bio-inspired spider silk webs under infrared and humidity stimulations, respectively. The response rate of the stimulated spider webs under infrared conditions was significantly better than that under humidity stimulation. However, the final shrinkage and deformation of the bio-inspired spider webs were similar, and it can be concluded that the different stimulation methods affected the shrinkage and deformation rates of the bio-inspired spider webs. However, the final shrinkage and deformation of the bio-inspired spider webs will not be affected if the stimulation time is sufficient, which is mainly determined by the state of the bio-inspired spider webs before this stimulation.

The woven bio-inspired spider web can shrink and deform under infrared or humidity stimulation. The degree of deformation increases with an increase in stimulation time, and the rate of shrinkage and deformation of the bio-inspired spider silk web varies under different stimulation methods. Thus, by controlling the manner, degree and timing of stimulation, bio-inspired spider silk webs with different degrees of contraction and deformation can be obtained. Furthermore, the observed dynamic changes in the shape and mechanical properties of bio-inspired spider webs can be applied in different scenarios.

### 3.5. Application Experiments

Figure 9a,b show the dynamic response of a bio-inspired spider web to the impact of different objects. Bio-inspired spider webs are formed by sequentially interweaving biologically inspired spider silk with a double predetermined structure, as described in the previous section. Figure 9(a_1_–a_9_) show the deformation of the bio-inspired spider web in response to the impact of a small glass ball; Figure 9(b_1_–b_9_) show the deformation of the bio-inspired spider web in response to the impact of a larger tennis ball. The entire process is shown by the forced deformation when resisting the impact, the accumulation of force to rebound, and the ball bouncing back. After each rebound, the kinetic energy of the ball is absorbed by the bio-inspired spider web, and the velocity gradually decreases.

Figure 9c shows the concealment of spider webs woven with bio-inspired spider silk with different levels of stimulation. The web woven by the sample that underwent only one high-temperature stimulation (Figure 9(c_1_)) had better concealment under this condition and was difficult to detect. Figure 9(c_2_) shows that, with three high-temperature stimulations, the bio-inspired spider silk had the second highest concealment; Figure 9(c_3_) shows that, with five high-temperature stimulations, the bio-inspired spider silk had the worst concealment and could be detected more easily by observation.

As the number of stimulations increased, the contraction and spiral deformation of the bio-inspired spider silk increased, and the concealment and mechanical properties of the bio-inspired spider web dynamically changed. The concealment of the bio-inspired spider web worsened. However, the mechanical properties of the samples improved to a certain extent. Thus, a balance in practical applications must be found to achieve the desired effect. This type of bio-inspired spider web can be used for stealth interception. Generally, when the number of stimulations is lower, the concealment is better. When necessary, the number of stimulations can be rapidly increased to improve the impact resistance of the bio-inspired spider silk web for intercepting impact objects. It can be applied in underwater interception, aerospace, military bases, and many other fields with special requirements.

The bio-inspired spider web made of polyurethane can produce different dynamic deformation with the stimulation of various temperatures. When the stimulation conditions are the same, the dynamic change process can be controlled by changing the density and size of the nodes. The existing research on bio-inspired spider webs mainly focuses on generating artificial bio-inspired fibers with mechanical properties similar to those of natural spider silk, or investigating the mechanical properties exhibited by spider web structures in response to different types of impacts [30]. We took two approaches to control the dynamics of the mechanical properties of bio-inspired spider webs via pre-programming and varying the degrees of stimulation, which successfully achieved a certain degree of controlled dynamic changes in the bio-inspired spider webs.

## 4. Conclusions

The effects of various stimulation methods on the mechanical properties of 4D-printed bio-inspired spider silk were investigated and analyzed. The results demonstrated that the tensile properties of the bio-inspired spider silk fabricated using the 4D printing method in this study can be dynamically varied using stimulation methods such as temperature, humidity, and infrared light. The essence of this method is that by stimulating the bio-inspired spider silk material, it can be contracted and deformed to form a relatively uniform spiral structure in a localized area. By varying the stimulation method and time, the size and amount of the spiral region formed by the contraction and deformation can be controlled. This controls the shape and size of the energy-absorbing unit formed by the bio-inspired spider silk after stimulation, which further affects the mechanical properties of the bio-inspired spider silk. The energy-absorbing units can be formed not only according to the stimulation method, but also in the form of nodes that are preset into the printed bio-inspired spider silk structure. This allows the formation of a part of the energy-absorbing unit in the form of an advanced burial. Through the double preset coupling of these two types of energy-absorbing units, the proportion and size of each part of the energy-absorbing unit can be controlled so that the mechanical properties of the bio-inspired spider silk, particularly the tensile properties, can be dynamically changed. Controlling the dynamics of bio-inspired spider silk further enables dynamic changes in the impact resistance of the woven bio-inspired spider silk web. We have proposed a relatively simple way to control the dynamic changes in the mechanical properties of bio-inspired spider silk, resulting in a dynamic 4D-printed bio-inspired spider web. This can lead to the further development of application areas involving bio-inspired spider webs.

## Figures and Tables

**Figure 1 polymers-14-02069-f001:**
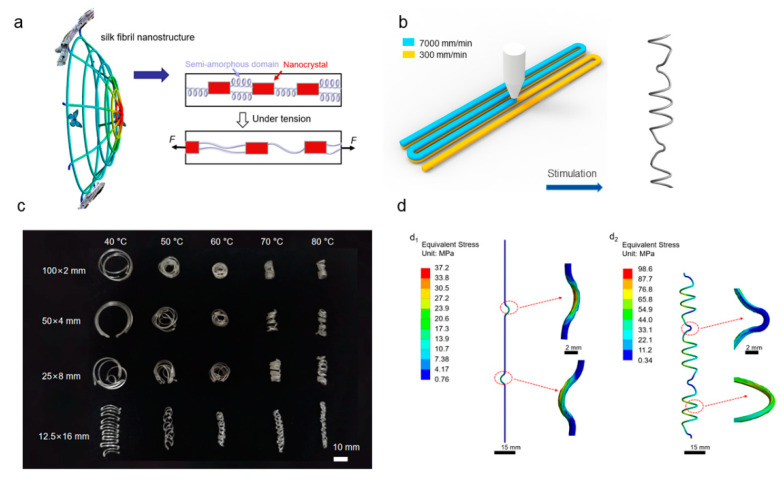
Schematic of bio-inspired spider webs catching prey when impacted (**a**); printing principle demonstration (**b**); bio-inspired spider silk is deformed with various nodes by stimulated shrinkage (**c**); simulation analysis of stress applied on two types of bio-inspired spider silk (**d**).

**Figure 2 polymers-14-02069-f002:**
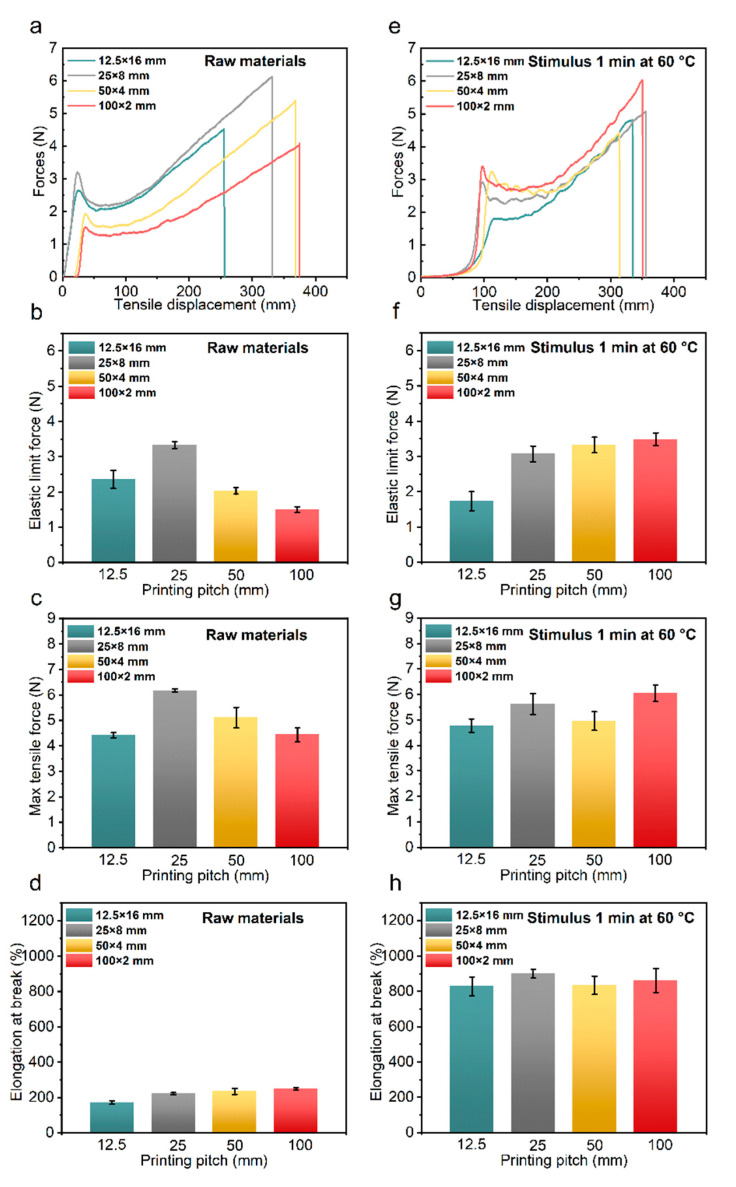
Force and tensile displacement curves of the unstimulated bio-inspired spider silk samples with different nodes (**a**); elastic limit force (**b**); maximum tensile force (**c**); elongation at break (**d**). Force versus tensile displacement curves of bio-inspired spider silk samples with different nodes stimulated at 60 °C for 1 min (**e**); elastic limit force (**f**); maximum tensile force (**g**); elongation at break (**h**).

**Figure 3 polymers-14-02069-f003:**
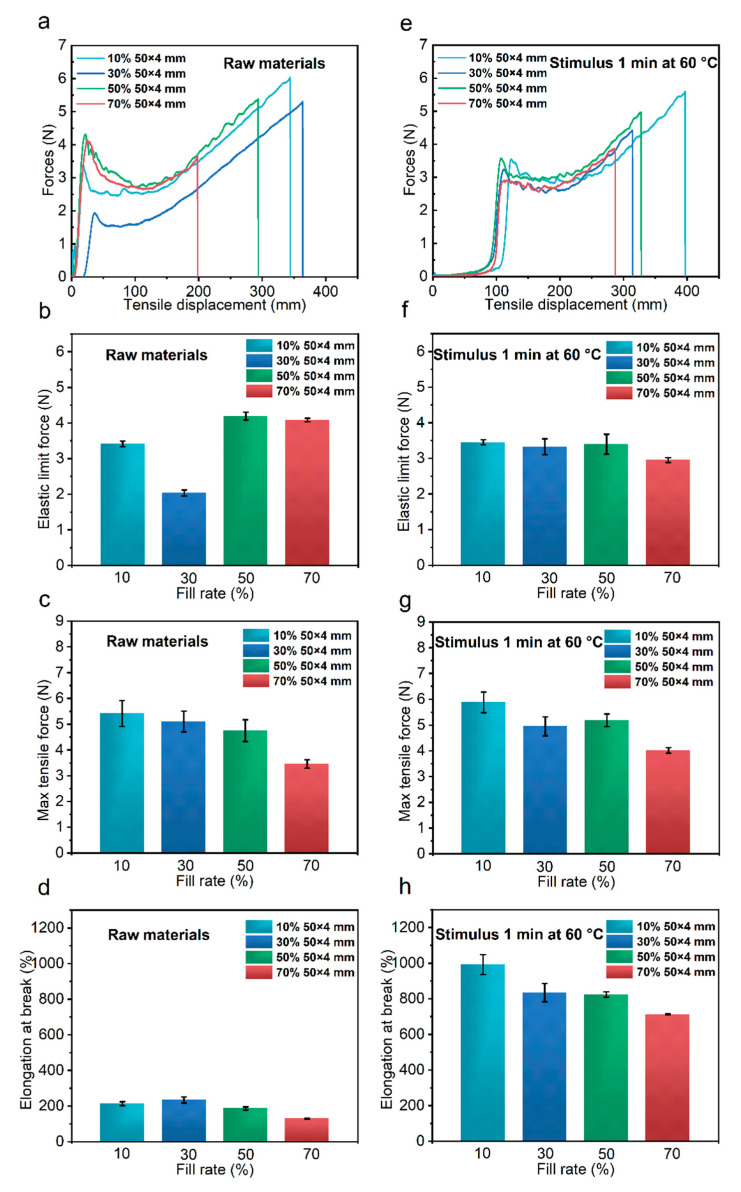
Force versus tensile displacement curves of the unstimulated bio-inspired spider silk samples with different filling densities (**a**); elastic limit force (**b**); maximum tensile force (**c**); elongation at break (**d**). Force versus tensile displacement curves of the bio-inspired spider silk samples with different filling densities stimulated for 1 min at 60 °C (**e**); elastic limit force (**f**); maximum tensile force (**g**); elongation at break (**h**).

**Figure 4 polymers-14-02069-f004:**
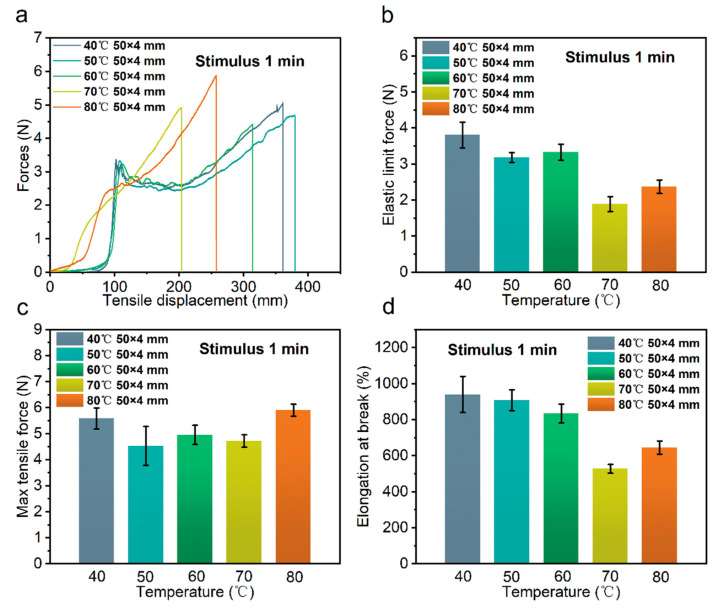
Force versus tensile displacement curves of the bio-inspired spider materials stimulated for 1 min at different temperatures (**a**); elastic limit force (**b**); maximum tensile force (**c**); elongation at break (**d**).

**Figure 5 polymers-14-02069-f005:**
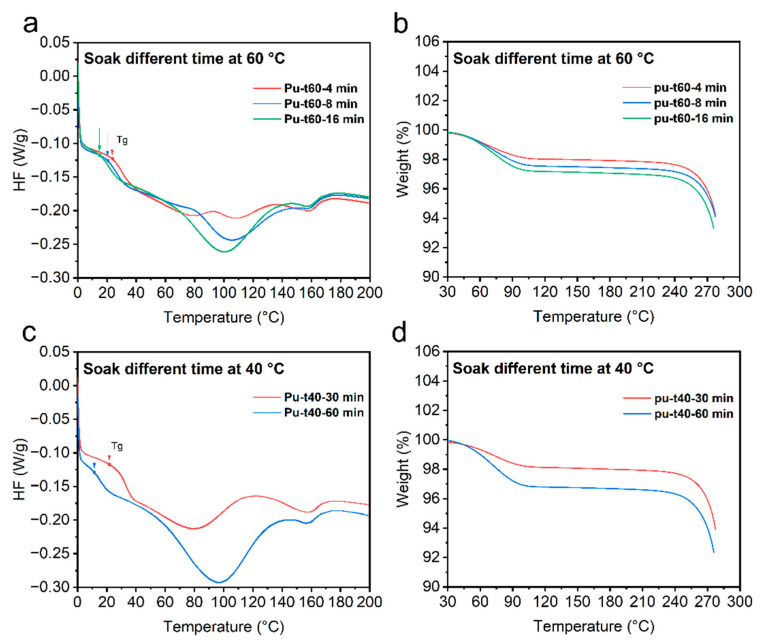
DCS experiments of bio-inspired spider silk samples soaked at 60 °C (**a**); TGA experiments of bio-inspired spider silk samples soaked at 60 °C (**b**); DCS experiments of bio-inspired spider silk samples soaked at 40 °C (**c**); TGA experiments of bio-inspired spider silk samples soaked at 40 °C (**d**).

**Figure 6 polymers-14-02069-f006:**
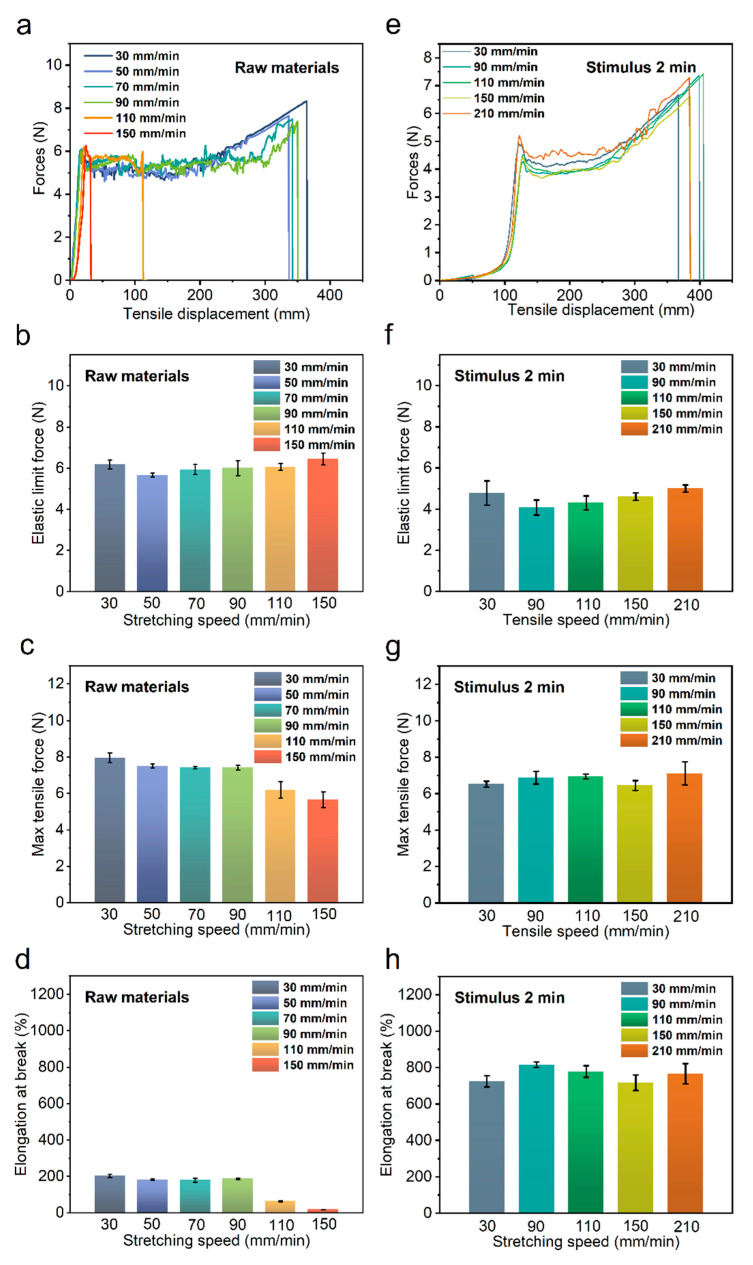
Force versus tensile displacement curves of unstimulated raw bio-inspired spider silk samples at different tensile speeds (**a**); elastic limit force (**b**); maximum tensile force (**c**); elongation at break (**d**). Force versus tensile displacement curves for bio-inspired spider material stimulated at 60 °C for 2 min at different tensile speeds (**e**); elastic limit force (**f**); maximum tensile force (**g**); elongation at break (**h**).

**Figure 7 polymers-14-02069-f007:**
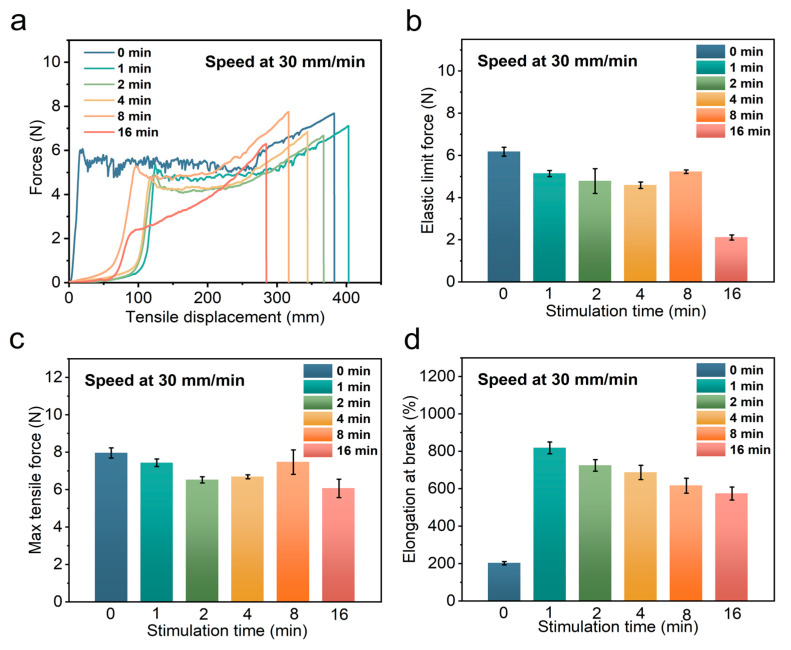
Force versus tensile displacement curves for bio-inspired spider material stimulated at 60 °C for different times (**a**); elastic limit force (**b**); maximum tensile force (**c**); elongation at break (**d**).

**Figure 8 polymers-14-02069-f008:**
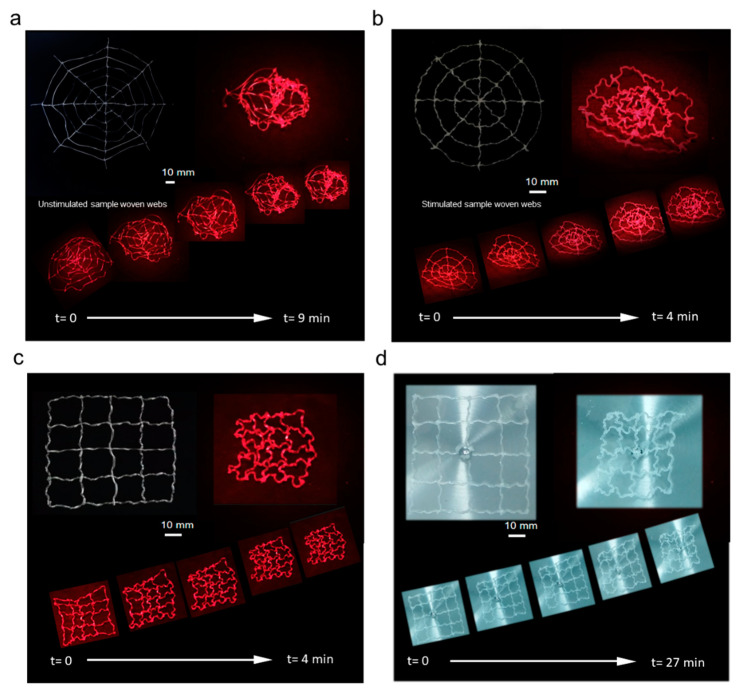
Stimulated deformation of the unstimulated spider webs made of bio-inspired spider silk (**a**); stimulated deformation of the webs made of bio-inspired spider silk stimulated at high temperature (**b**); stimulated deformation of square bio-inspired spider webs stimulated by infrared light (**c**); stimulated deformation of square bio-inspired spider webs stimulated by humidity (**d**).

**Figure 9 polymers-14-02069-f009:**
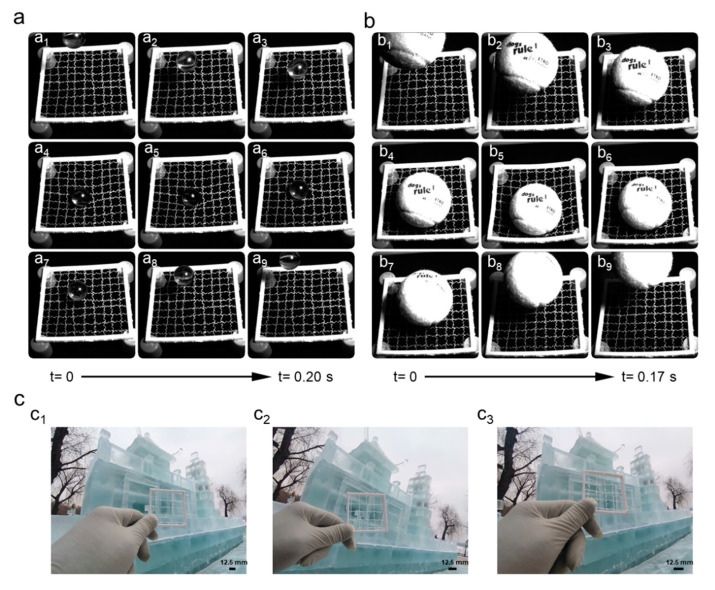
The glass ball impacts the bio-inspired spider silk web (**a**); tennis ball impacts the bio-inspired spider silk web (**b**); the concealment of the web woven by bio-inspired spider silk with different levels of stimulation (**c**).

## Data Availability

The data presented in this study are contained within the article.

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
