# Peer review of "Bio-Inspired 4D Printing of Dynamic Spider Silks"

_polymers, 2022, doi:10.3390/polym14102069_

Round 1

Reviewer 1 Report

In this paper, the developers use 4D printing technology to make the bio-inspired spider silk obtain the elasticity and rigidity of spider silk by burying nodes during the printing process. The design of the preset nodes and the helical deformation of the material as the preset energy absorption unit is very attractive. In this paper, the dynamic research of bionic spider silk is carried out through stimulation methods such as temperature, humidity, and infrared light, and there are abundant data for readers' reference. This article is informative and interesting, but there are still some issues to be aware of.

  1. There is a repetition of a period at the end of 200, please delete the redundant symbols. And there is a missing period at 279, please check the text carefully for the related symbol errors.
  2. It is suggested to change the description of the number of nodes to the density of nodes, which seems more appropriate.
  3. You describe the changes in the tensile properties of bio-inspired spider silk under different temperature stimuli, but do not discuss in depth. At the same time, many sets of data in Figure 2 have rich and interesting irregular changes, which will be valuable if more in-depth discuss can be carried out.
  4. The typesetting of pictures (for example: Figure 3 and Figure 2) is not conducive to readers' contacting the picture information during reading. It is recommended to split the pictures so that each picture can be closer to the text content it associates.
  5. In the ball impact test, the impact mode of the sphere should also be informed, such as: free fall, fall height, launch force, etc.
  6. In Figure 5.C, it seems more convincing whether it is possible to use a fixed light source, a fixed background, and a fixed angle for shooting.
  7. In the analysis of multiple sets of data, the author only carried out a descriptive analysis of the data changes. It is hoped that the author can carry out related research on the molecular level changes and mechanisms of materials in the follow-up research, which may be more valuable information.

Author Response

Point 1: There is a repetition of a period at the end of 200, please delete the redundant symbols. And there is a missing period at 279, please check the text carefully for the related symbol errors.

Response 1: Thanks for the reviewer’s suggestion. We have revised the errors you raised, and checked again to make sure there were no relevant symbolic errors in the text.

Point 2: It is suggested to change the description of the number of nodes to the density of nodes, which seems more appropriate.

Response 2: Thanks for the reviewer’s suggestion. We have transformed the description of the number of nodes into the density of nodes.

Point 3: You describe the changes in the tensile properties of bio-inspired spider silk under different temperature stimuli, but do not discuss in depth. At the same time, many sets of data in Figure 2 have rich and interesting irregular changes, which will be valuable if more in-depth discuss can be carried out.

Response 3: Thanks for the reviewer’s suggestion. We have added some more detailed discussion and analysis as follows.

The size and density of energy-absorbing units affect the stress distribution of the bio-inspired spider silk sample during stretching, which affect the dynamics of the mechanical properties of the sample under different conditions. Two types of energy absorption units exist in the samples. The density and size of the nodes are preset when printing the bio-inspired spider silk sample as the preset energy-absorbing unit. The spiral deformation of the sample after stimulation can be regarded as a large node as the additional energy absorption unit. The energy-absorbing unit is mainly controlled by temperature during stimulation, which can control the degree of spiral deformation. The influences of the density and size of nodes, and stimulation temperature on the tensile properties of bio-inspired spider silk samples were investigated.

We synthesized the effects of node density and size as well as stimulation temperature on the tensile properties of bio-inspired spider silk samples. Different stimulation temperatures influenced the differences in the large nodes formed with the bio-inspired spider silk samples after helical deformation. At the nodes, the abrupt change in shape produced the stress concentration phenomenon. Differences in the type, density and size of the nodes produced differences in the stress concentration, which further influenced the bio-inspired spider silk tensile properties.

Point 4: The typesetting of pictures (for example: Figure 3 and Figure 2) is not conducive to readers' contacting the picture information during reading. It is recommended to split the pictures so that each picture can be closer to the text content it associates.

Response 4: Thanks for the reviewer’s suggestion. We have reformatted these pictures (for example: Figure 3 and Figure 2). The Figure 2 was splitted into Figure 2, 3, 4. The Figure 3 was splitted into Figure 5, 6, 7. The relevant pictures have been further splitted up to make the discussion more connected to the pictures.

Point 5: In the ball impact test, the impact mode of the sphere should also be informed, such as: free fall, fall height, launch force, etc.

Response 5: Thanks for the reviewer’s suggestion. We have added the relevant description and revised this paragraph as follows.

Application experiments were also conducted, including an impact resistance experiment. The bio-inspired spider webs 100 × 100 mm in size were spaced 12.5 mm apart, and the impact experiments were conducted using a glass ball and tennis ball. During the impact test, it was ensured that the test was conducted under windless conditions. The vertical distance between the center of the ball and the net surface was 30 mm for both balls. No additional force was attached to the ball during its fall, and the impact was taken in a free-fall manner. The diameter of the glass balls was 25 mm, and the weight of the tennis balls was 55.638 g. Photographs were taken using a industrial camera (OSG030-815UM, Shenzhen Yingshi Technology Co., Ltd., Shenzhen, China) equipped with a 2 million 1/2" target surface industrial lens (HF-Z0412, Shen-zhen Yingshi Technology Co., Ltd. , Shenzhen, China) to demonstrate the feasibility of the bio-inspired spider silk applied to stealth interception.. The dynamic bio-inspired spider web was used to demonstrate the feasibility of intercepting impacts.

Point 6: In Figure 5.C, it seems more convincing whether it is possible to use a fixed light source, a fixed background, and a fixed angle for shooting.

Response 6: Thanks for the reviewer’s suggestion. We have considered your proposal very carefully. However, we are sincerely sorry for that we conducted this application experiment in winter time. And we have tried our best to make sure the pictures have similar background. At this stage, our lab has been closed due to the COVID-19, we are sincerely sorry for that we couldn’t take pictures again. We will be sincerely appreciated if you agree to the current version being published.

Point 7: In the analysis of multiple sets of data, the author only carried out a descriptive analysis of the data changes. It is hoped that the author can carry out related research on the molecular level changes and mechanisms of materials in the follow-up research, which may be more valuable information

Response 7: Thanks for the reviewer’s suggestion. We have added the relevant analysis below. Your reminder is extremely valuable to us, and we will further refine our analysis for these phenomena in the next experimental studies.

The effects of water on the internal molecular of polyurethane were investigated via the DSC and TGA experiments. The glass transition temperature of polyurethane decreases with the increase of water absorption [27]. The absorbed water can interact with the polymer, thus increasing the mobility of the polymer chains and leading to a decrease in the glass transition temperature [28].

Reviewer 2 Report

In the manuscript “Bio-inspired 4D Printing of Dynamic Spider Silks” the authors report the 3d printing of polyurethane structures, resembling the 3d structure of a spider web, and exploit the shape memory effect of the polymer to induce a shape change in the printed structures. Overall, the study is well conducted and the topic seems to be relevant to Polymers. However, there are some major points to be fixed to corroborate the authors’ claims. The detailed comments are as follows:

  • It is not clear, why you talk about "bio-inspired spider silk". Maybe what you mean is "bio-inspired spider web". What you are mimicking is the 3d architecture and performance of spider webs, not the chemical properties of spider silk. Please carefully check the manuscript to fix this issue.
  • The introduction should be revised, including an overview of other studies, present in the literature, mimicking the 3d architecture of spider webs.
  • In the results and discussion section, a critical discussion is missing. The discussion should introduce a comparison with the literature, which should be aimed primarily at highlighting the potential and innovation of the work. Moreover, this section lacks references.

Author Response

Point 1: It is not clear, why you talk about "bio-inspired spider silk". Maybe what you mean is "bio-inspired spider web". What you are mimicking is the 3d architecture and performance of spider webs, not the chemical properties of spider silk. Please carefully check the manuscript to fix this issue.

Response 1: Thanks for the reviewer’s suggestion. We have changed the term "bio-inspired spider silk" to "bio-inspired spider web", which is more relevant to the article. The specific modifications are as follows.

Title: Bio-inspired 4D Printing of Dynamic Spider silks Web

Point 2: The introduction should be revised, including an overview of other studies, present in the literature, mimicking the 3d architecture of spider webs.

Response 2: Thanks for the reviewer’s suggestion. We have added the relevant description and revised the Introduction section as follows.

Numerous studies have focused on the manufacture of bio-inspired spider silks. Vendrely et al., through genetic engineering methods, employed two biotechnological production strategies to successfully apply spider silk protein for production of spider silk [15]. Bhattacharyya et al. genetically engineered spider silk. Spider silk protein was synthesized in Escherichia coli after studying its structure and function, and a biologi-cal material similar to natural spider silk was produced, which advanced the produc-tion of synthetic spider silk[16]. Venkatesan et al. fabricated engineered major ampul-late spidroin 2 (eMaSp2) fibre by introducing foreign N- and C-terminal domains, and demonstrated that the fiber has good shape memory effects triggered by humidity fac-tors and the ability to restore stress [16]. Inspired by the microstructure of spider silk, Chan et al. effectively utilized the localised β-sheet domains  in the amorphous net-work considering the shortcomings of existing processing technology and synthesized a super composite material with a spider silk-like “amorphous/β-folding” microstruc-ture [17]. Dou et al. self-assembled homemade hydrogel fibers to create artificial spider silk. This faux spider silk fiber consists of a plastic sheath placed around an elastic core, which results in spider silk-like strength and stretchability [18]. When used in impact reduction applications, the bio-inspired spider silk fiber has a negligible rebound, allowing it to be applied to kinetic energy cushioning and shock reduction. Zhou et al. successfully designed a transparent composite material with super-impact resistance using the SBHL strategy. The biologically inspired spider webs are able to dissipate a significant amount of energy as they absorb the impact, and grab the projectile like a natural spider web [19]. Qin et al. created spider-web mimics com-posed of elastic thin filaments, and investigated the mechanical response of the elastic web under a variety of loading conditions [20].

Point 3: In the results and discussion section, a critical discussion is missing. The discussion should introduce a comparison with the literature, which should be aimed primarily at highlighting the potential and innovation of the work. Moreover, this section lacks references.

Response 3: Thanks for the reviewer’s suggestion. We have added the following relevant discussions.

The bio-inspired spider web made of polyurethane can produce different dynamic deformation with the stimulation of arious temperatures. When the stimulation conditions are the same, the dynamic change process can be controlled by changing the density and size of the nodes. The existing researches on bio-inspired spider webs mainly focused on generating artificial bio-inspired fibers with mechanical properties similar to those of natural spider silk, or investigating the mechanical properties exhibited by spider web structures in response to different types of impacts[30]. We took two approaches to control the dynamics of the mechanical properties of bio-inspired spider web via pre-programming and varying the degrees of stimulation, which successfully achieved the certain degree of controlled dynamic changes in the bio-inspired spider web.

Round 2

Reviewer 1 Report

Accept

Author Response

The authors sincerely thanks for the reviewer's guidance.

Reviewer 2 Report

The authors have not responded to Point 3. As already stated, a critical discussion is missing in this work. The authors simply pledged to add a new paragraph, but a discussion is lacking throughout the results and discussion section. 

This way, the manuscript looks more like a technical report than a scientific paper. 

I invite the authors to make a substantial change of the manuscript following the above-mentioned comments. These modifications are essential for improving the scientific quality of the work and meet the journal's standards.

Author Response

Response to Reviewer Comments

Point 1: Comments and Suggestions for Authors

The authors have not responded to Point 3. As already stated, a critical discussion is missing in this work. The authors simply pledged to add a new paragraph, but a discussion is lacking throughout the results and discussion section. 

This way, the manuscript looks more like a technical report than a scientific paper. 

I invite the authors to make a substantial change of the manuscript following the above-mentioned comments. These modifications are essential for improving the scientific quality of the work and meet the journal's standards.

Response 1: The authors sincerely thanks for the reviewer’s suggestion. We have added some new paragraphs and more critical discussions to the " Results and Discussion " section. We will be sincerely appreciated if you agree to the current version being published. The specific changes are listed as follows.

3.1 Bio-inspired 4D printing of dynamic spider silks

We took the form of preparing bio-inspired spider silk with different nodes by pre-programming the process parameters. As shown in Figure 1(c), the higher the temperature during stimulating, the more pronounced the spiral bending phenomenon of the bio-inspired spider silk. On the basis of the original bending of the nodes being retained, the linear part of the printed bio-inspired spider silk also undergoes spiral deformation. And when the bio-inspired spider silk is subjected to tensile deformation, the spiral deformation region generated by the linear part will be the first to be subjected to a larger value of equivalent force and unfold to resist the tensile impact. In this way, together with the bending at the original nodes, a double pre-setting is achieved to enhance the tensile impact resistance of the bio-inspired spider silk. This method uses a double-preset approach to provide a better energy-absorption effect.

3.2 Effect of energy-absorbing units on the tensile properties of bio-inspired spider silks

3.2.1. Effect of node density

The unstimulated biomimetic silk has only the first type of energy absorbing units with the preset nodes. When the sample is subjected to tension, the nodes are subjected to a large value of equivalent force. However, the stress concentration at the nodes also af-fects the linear part. When the linear section is short (e.g., 12.5 mm), the elastic tensile limit and the maximum tensile force of the unstimulated sample will be reduced. Therefore, adopting appropriate nodal densities (e.g., 50*4 and 25*8) can improve the tensile me-chanical properties of bio-inspired spider silk.

We further investigated the differences in the tensile properties of the stimulated bio-inspired spider silk, as shown in Figure 2(e-h). The elongation at break of the bio-inspired spiders with different densities of nodes after stimulation remained the same, and there was no significant difference in the maximum tensile force value; only the elastic limit force value increased with a reduction in the density of nodes of the raw bio-inspired spider material. This is because a smaller density of nodes results in a longer linear segment, as shown in Figure 1(d2), and the equivalent maximum stress in the bio-inspired spider silk after stimulation when it is stretched by force occurs at the bend formed by contraction of the linear segment after stimulation. In contrast, the 100*2 bio-inspired spider silk had the longest original straight segment and the most bending spirals formed at the same conditioned stimulation; thus, the elastic limit force of the bio-inspired spider silk with the lowest density of these node became the largest. This shows that the difference in the density of nodes causes a significant difference in the tensile properties of bio-inspired spider silks.

Two types of energy absorbing units are present in the stimulated bio-inspired spider silk sample. The spiral curved area formed by the stimulated bioinspired spider silk sample acts as the second type of energy absorbing unit and is the first to be stretched by the force. Bio-inspired spider silks with different node densities have different numbers of the first type of energy absorption units: the preset nodes, and they have different lengths of linear regions, which form the second type of energy absorption units. The combined effect of the two types of energy-absorbing units leads to significant differences in the tensile properties of the stimulated bio-inspired spider silk samples.

3.2.2. Effect of node size

In the previous section we showed that the densities of nodes significantly affected the tensile properties of bio-inspired spider silk, and the size of the nodes should also affect the tensile properties of bio-inspired spider silk. The filling density parameter during printing was set to 10%, 30%, 50%, and 70% to obtain four types of bio-inspired spider silk structures with different node sizes. Among them, the bio-inspired spider silk node was the largest at the 10% filling density, and the smallest at the 70% filling density. As shown in Figure 3(a-d), tensile experiments were performed on unstimulated raw bio-inspired spider silk materials with different filling densities. The nodes of the samples printed at 70% filling density were the smallest, and extremely small nodes lead to more obvious stress concentration, which causes the bio-inspired spider silk to fracture in the middle of the stretching; the elongation at breakage of this bio-inspired silk sample is the smallest. The maximum tensile force of the unstimulated biomimetic 4D printed bio-inspired spider silk sample at this point decreases with the size of the nodes, as shown in Figure 3(c), which could also be because smaller nodes lead to a more pronounced stress concentration phenomenon, which in turn reduces the maximum tensile force. Next, the raw bio-inspired spider silk material with different filling densities was stimulated and subjected to tensile experiments, as shown in Figure 3(e-h). At this point, the bio-inspired spider silk has two types of energy-absorbing units after stimulation. The analysis performed under this condition is shown in Figure 3(f); only the elastic limit force of the bio-inspired spider silk sample with 70% filling density is slightly smaller, which may be caused by the small nodes and obvious stress concentration. The fracture tensile rate of the bio-inspired spider silk sample under this condition decreased with a reduction in node size; the value of elongation at break of the sample printed at 70% filling density was also the smallest.

The size of nodes also significantly affects the tensile properties of bio-inspired spider silk sample. When the nodes are small, the stress concentration is extremely pronounced and the degree of bending at this node is extremely dramatic. This results in that the smaller the node, the more likely it is to fracture due to excessive stress concentration. This results in relatively smaller values of maximum tensile force and fracture elongation for bio-inspired spider silk samples with smaller nodes.

3.2.3. Effect of stimulation temperature

Finally, the effect of stimulation temperature on the tensile properties of the bio-inspired spider silk samples was investigated, as shown in Figure 4(a-d). The elastic limit force, maximum tensile force, and elongation at break of the samples were not significantly different when the stimulation temperature was relatively low, e.g., at 40, 50, and 60°C, for the constant-temperature water stimulation. When the temperature was again increased to 70°C, the elastic limit force and elongation at break of the stimulated bio-inspired spider samples significantly reduced, while their maximum tensile force magnitude remained essentially unchanged. When the temperature was raised to 80°C again, the elastic limit force, maximum tensile force, and elongation at break increased compared with those at 70°C. The elastic limit force and elongation at break were lower than those of the bio-inspired spider samples stimulated by constant-temperature water at 40, 50, and 60°C, but the maximum tensile force remained the same. The influence of the stimulation temperature on the tensile properties of bio-inspired spider samples cannot be ignored.

We synthesized the effects of node density and size as well as stimulation temperature on the tensile properties of bio-inspired spider silk samples. Different stimulation temperatures influenced the differences in the large nodes formed with the bio-inspired spider silk samples after helical deformation. At the nodes, the abrupt change in shape produced the stress concentration phenomenon. Differences in the type, density and size of the nodes produced differences in the stress concentration, which further influenced the bio-inspired spider silk tensile properties.

3.3 Effect of tensile speed and moisture absorption content on the tensile properties of bio-inspired spider silks.

 3.3.1. The DSC and TGA results

The effects of water on the internal molecular of polyurethane were investigated via the DSC and TGA experiments. The glass transition temperature of polyurethane de-creases with the increase of water absorption [27]. The absorbed water can interact with the polymer, thus increasing the mobility of the polymer chains and leading to a decrease in the glass transition temperature [28].

3.3.2. Effect of different stretching speeds

The unstimulated bio-inspired spider silk samples were subjected to larger tensile speeds, which resulted in sudden breakage of the bio-inspired spider silk samples during stretching, causing a significant decrease in the fracture elongation of the samples at the higher tensile speeds. In order to avoid mid-rupture, the sample can be soaked to increase the relative humidity of the sample, so that the tensile speed is increased from the value that caused the unstimulated spider silk sample to break, and the stimulated sample still does not break mid-rupture. Thus, by soaking and humidifying the samples, breakage in the middle of the process at a high stretching speed can be prevented.

3.3.3. Effect of the degree of moisture absorption

When the stimulated bio-inspired spider silk samples were immersed for longer pe-riods of time, the elastic ultimate force and elongation at break decreased as the degree of moisture absorption of the bio-inspired spider silk samples increased. The tensile strength of the polymer decreased with increasing water absorption. This is related to the increase in toughness of the bio-inspired spider samples after a large increase in the degree of moisture absorption of the bio-inspired spider silk.

3.4 Bio-inspired spiderwebs stimulation experiment

The woven bio-inspired spider web can shrink and deform under infrared or humid-ity stimulation. The degree of deformation increases with an increase in stimulation time, and the rate of shrinkage and deformation of the bio-inspired spider silk web varies under different stimulation methods. Thus, by controlling the manner, degree and timing of stimulation, bio-inspired spider silk webs with different degrees of contraction and de-formation can be obtained. Furthermore, the observed dynamic changes in the shape and mechanical properties of bio-inspired spider webs can be applied in different scenarios.

3.5 Application experiments

As the number of stimulations increased, the contraction and spiral deformation of the bio-inspired spider silk increased, and the concealment and mechanical properties of the bio-inspired spider web dynamically changed. The concealment of the bio-inspired spider web worsened. However, the mechanical properties of the samples improved to a certain extent. Thus, a balance in practical applications must be found to achieve the de-sired effect. This type of bio-inspired spider web can be used for stealth interception. Gen-erally, when the number of stimulations is lower, the concealment is better. When neces-sary, the number of stimulations can be rapidly increased to improve the impact re-sistance of the bio-inspired spider silk web for intercepting impact objects. It can be ap-plied in underwater interception, aerospace, military bases, and many other fields with special requirements.

The bio-inspired spider web made of polyurethane can produce different dynamic deformation with the stimulation of various temperatures. When the stimulation condi-tions are the same, the dynamic change process can be controlled by changing the density and size of the nodes. The existing researches on bio-inspired spider webs mainly focused on generating artificial bio-inspired fibers with mechanical properties similar to those of natural spider silk, or investigating the mechanical properties exhibited by spider web structures in response to different types of impacts[30]. We took two approaches to control the dynamics of the mechanical properties of bio-inspired spider web via pre-programming and varying the degrees of stimulation, which successfully achieved the certain degree of controlled dynamic changes in the bio-inspired spider web.

Round 3

Reviewer 2 Report

The manuscript can be accepted for publication